# LLaVA-Video: Video Instruction Tuning With Synthetic Data

**Yuanhan Zhang**                                                   *yuanhan002@e.ntu.edu.sg*
*S-Lab, Nanyang Technological University*

**Jinming Wu**                                                        *wjm_18@bupt.edu.cn*
*BUPT*

**Wei Li**                                                       *liwei.speech@bytedance.com*
*ByteDance*

**Bo Li**                                                           *libo0013@e.ntu.edu.sg*
*S-Lab, Nanyang Technological University*

**Zejun Ma**                                                        *mazejun@bytedance.com*
*ByteDance*

**Ziwei Liu**                                                         *ziwei.liu@ntu.edu.sg*
*S-Lab, Nanyang Technological University*

**Chunyuan Li**                                                   *lichunyuan24@gmail.com*
*ByteDance*

**Reviewed on OpenReview:** *https://openreview.net/forum?id=EElFGvt39K*

## Abstract

The development of video large multimodal models (LMMs) has been hindered by the difficulty of curating large amounts of high-quality raw data from the web. To address this, we consider an alternative approach, creating a high-quality synthetic dataset specifically for video instruction-following, namely LLaVA-Video-178K. This dataset includes key tasks such as detailed captioning, open-ended question-answering (QA), and multiple-choice QA. By training on this proposed dataset, in combination with existing visual instruction tuning data, we introduce LLaVA-Video, a new video LMM. Our experiments demonstrate that LLaVA-Video achieves strong performance across various video benchmarks, highlighting the effectiveness of our dataset. We plan to release the dataset, its generation pipeline, and the model checkpoints.

## 1 Introduction

We are in an era where large-scale computing and data is crucial for multimodal learning (Li et al., 2024d). A significant recent advancement was introduced by visual instruction tuning (Liu et al., 2024a), which laid the foundation for building a general-purpose visual assistant. Notably, it proposed a data generation pipeline to create high-quality image-language instruction-following data. This pipeline has inspired subsequent researches (Li et al., 2024c;b;a; Lin et al., 2024) aimed at generating diverse image-language instruction data across various visual domains, accelerating the development of visual instruction tuning techniques.

Compared to the construction of image-language instruction-following data, obtaining high-quality video-language instruction-following data is challenging (Zhang et al., 2023; Li et al., 2024e). First, sourcing high-quality videos is difficult. We need to find videos with significant temporal changes that provide more knowledge than what image-language data can offer. However, we have found that most videos in current video-language instruction-following datasets (Chen et al., 2024a; Zhang et al., 2024d) are relatively

static. Additionally, these videos are mostly trimmed based on scene changes, resulting in simplified plots. Such simplified video-language instruction-tuning data is inadequate for models to understand videos with complex narratives. Furthermore, current video-language instruction-following datasets often use a very sparse sampling rate for frame annotation. For instance, ShareGPT4Video (Chen et al., 2024a) has an average sampling rate of 0.15, sometimes sampling only 2 frames from a 30-second video. This sparse sampling rate is effective in describing overall scenes but fails to capture detailed movements or changes in the video, resulting in hallucination when detailed descriptions of the video are required.

To overcome these shortcomings, we introduce a comprehensive video instruction-tuning dataset named LLaVA-Video-178K, consisting of 178,510 videos ranging from 0 to 3 minutes. This dataset is enriched with detailed annotations, open-ended questions, and multiple-choice questions, developed through a combination of GPT-4o (OpenAI, 2024) and human efforts. It features four favorable properties: (*i*) **Extensive Video Source:** We conduct a comprehensive survey on the video sources of exsiting video understanding datasets, and conclude 10 major video data sources, from which we start our video data collection by building a video pool. Although there are over 40 video-language datasets, their video data are mainly sourced from 10 datasets (Zhou & Corso, 2017; Xue et al., 2022; Goyal et al., 2017; Caba Heilbron et al., 2015; Kay et al., 2017; Sigurdsson et al., 2016; Wang et al., 2023; Shang et al., 2019; Grauman et al., 2022; Zhu et al., 2023a), covering a wide range of video domains, such as activities, cooking, TV shows, and egocentric views. (*ii*) **Dynamic Untrimmed Video Selection:** From these sources, we use several filtering logic to select the most dynamic videos from the video data pool. Notably, we select original, untrimmed videos to ensure plot completeness. (*iii*) **Recurrent Detailed Caption Generation Pipeline with Dense Frame Sampling:** We propose a detailed video caption pipeline that operates recurrently, enabling us to generate detailed captions for videos of any length. This pipeline has three levels, each level of description represents a different time-range: from 10 seconds to the entire video length. It is recurrent as the historical description from any level serves as the context for generating new descriptions at any level. Additionally, we adopted a dense sampling strategy of one frame per second to ensure the sampled frames are rich enough to represent the videos. (*iv*) **Diverse Tasks:** Based on the detailed video descriptions, we can generate question-answer pairs. To ensure our questions cover a wide range of scenarios, by referring to the video question-answering dataset, we define 16 question types. We prompt GPT-4o to generate question-answer pairs by referring to these question types, covering open-ended and multi-choice questions.

Based upon the LLaVA-Video-178K dataset, we developed LLaVA-Video. Contrary to previous studies suggesting that training with single frames is sufficient for video-language understanding (Lei et al., 2022), our findings reveal a significant impact of frame count on LLaVA-Video's performance, attributable to the detailed features of LLaVA-Video-178K. Observing this, we explored maximizing frame sampling within the constraints of limited GPU memory. We introduce LLaVA-Video $_{\texttt{SlowFast}}$, a video representation technique that optimally distributes visual tokens across different frames. This approach allows for incorporating up to three times more frames than traditional methods, which allocate an equal number of visual tokens to each frame.

Our contributions are as follows:

- *Video-language Instruction-Following Data*: We present a high-quality dataset *LLaVA-Video-178K* tailored for video instruction-following. It consists of 178K video with 1.3M instruction samples, including detailed captions, free-form and multiple-choice question answering.
- *Video Large Multimodal Models*: We develop *LLaVA-Video*, a series of advanced large video-language models that expand the capabilities of open models in understanding video content.
- *Open-Source*: In an effort to support the development of general-purpose visual assistants, we release our multimodal instruction data, codebase, model checkpoints, and a visual chat demo to the public.

## 2    Related Work

In this work, our goal is to create a high-quality video-language dataset that goes beyond simple video captions. We aim to improve the ability to follow instructions, which includes detailed video descriptions, open-ended video question-answering, and multiple-choice video question-answering data. We discuss related datasets in

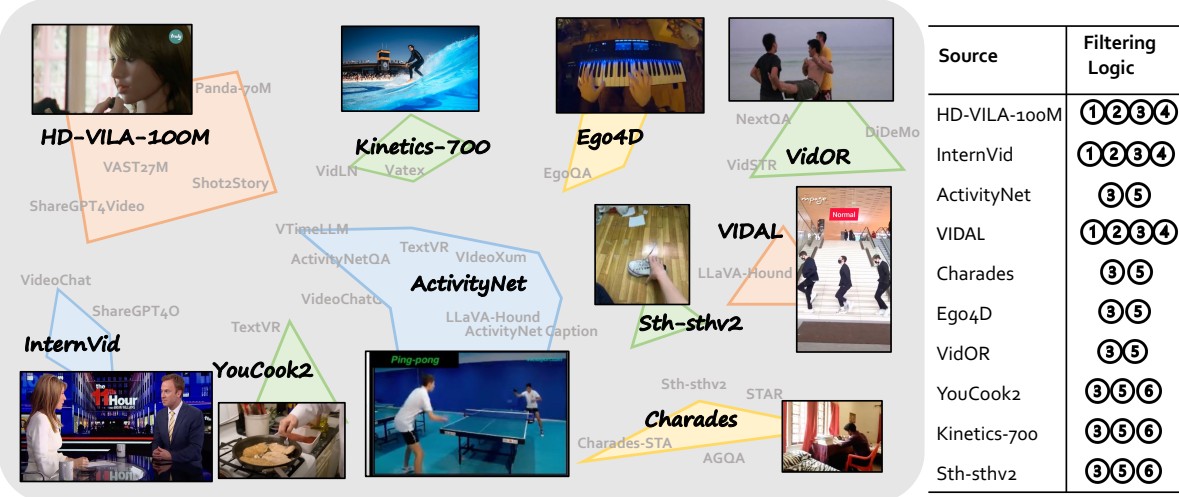

Figure 1: **Video sources in the proposed *LLaVA-Video-178K*.** (Left) The relationship between 10 video sources we have utilized and other existing video-language datasets. (Right) Filtering logic for video sources. The detail of filtering logic: ① Sorted by Views, ② Number of scenes greater than 2, ③ Video duration between 5 seconds and 180 seconds, ④ Ratio of scenes to video duration less than or equal to 0.5, ⑤ Resolution greater than 480p, ⑥ 50 samples for each category.

Table 1. Previous video-language datasets (Miech et al., 2019) include manually annotated data for various tasks, such as video captions (Chen & Dolan, 2011; Xu et al., 2016; Rohrbach et al., 2015; Anne Hendricks et al., 2017a; Caba Heilbron et al., 2015; Zhou & Corso, 2017), and video question-answering (Yu et al., 2019; Zadeh et al., 2019; Xiao et al., 2021). However, manual annotation is expensive and limits the size of such datasets. To address the shortage of data, studies like (Miech et al., 2019; Lee et al., 2021; Zellers et al., 2021; Xue et al., 2022) suggest automatically annotating data using subtitles created by ASR. While this method greatly expands the dataset size to 100 million samples, the subtitles often fail to accurately describe the main video content. Additionally, other studies (Xu et al., 2017; Grunde-McLaughlin et al., 2021; Wu et al., 2024a) use language models (Xu et al., 2017) or question templates (Grunde-McLaughlin et al., 2021; Wu et al., 2024a) to generate question-answer pairs. Although this approach can generate a large number of questions and answers, it often produces poor-quality questions that do not reflect real-world user inquiries. More recent research (Chen et al., 2024b) has prompted video-language models such as BLIP-2 (Li et al., 2023), VideoChat (Li et al., 2024e), Video-LLaMA (Zhang et al., 2023), and MiniGPT-4 (Zhu et al., 2023b) to generate video captions. However, these models are limited in their ability to provide detailed descriptions.

The most related works to ours are the recent AI-generated synthetic video instruction tuning data, Islam et al. (2024) introduced Video ReCap, which recursively annotates video captions. Unlike Video ReCap, each clip-wise (level-1) description in our pipeline is generated with historical context. This ensures that connections from previous events in the video timeline are linked to the current event. LLaVA-Hound (Zhang et al., 2024d) and ShareGPT4Video (Chen et al., 2024a), where they have used GPT-4 (OpenAI, 2023) to generate video captions and open-ended video question-answering. Although the quality of the captions and question-answer pairs has significantly improved, the video sources they use are too static to produce high-quality data for instruction-following scenarios. They also only use very sparse frames for prompting GPT-4V, which results in annotations that fail to capture nuanced actions and continuous plots in the videos. Additionally, Shot2Story (Han et al., 2023) and Vript (Han et al., 2023) also employ GPT-4V (OpenAI, 2023) for video captioning. Their outputs, however, include audio details, which are outside the scope of this study.

## 3 Video Instruction-Following Data Synthesis

A high-quality dataset for video instruction-tuning is crucial for developing effective video-language models. We identify a key factor in building such datasets: ensuring richness and diversity in both video content and

its language annotations. We perform comprehensive survey on the existing video benchmarks, covering across various public video captioning and question-answering datasets, then identify ten unique video sources that contribute to over 40 video-language benchmarks. From each source, we select videos that exhibit significant temporal dynamics. To maintain diversity in the annotations, we establish a pipeline capable of generating detailed captions for videos of any length. Additionally, we define 16 types of questions that guide GPT-4o in creating question-answer pairs to assess the perceptual and reasoning skills of the video-language models.

## 3.1 Video source

One important starting point in building a high-quality video instruction-following dataset is to find a sufficiently diverse pool of video data. From this pool, we can select the qualified videos. In our study of public video-language datasets—including video captioning, video question answering, video summarization, and moment-wise captioning—we noticed that although different datasets focus on various video understanding tasks (*e.g.,* , AGQA (Grunde-McLaughlin et al., 2021) for spatial-temporal relations and STAR (Wu et al., 2024a) for situational reasoning), most are sourced from ten main video sources. For instance, both AGQA and STAR use data from Charades (Sigurdsson et al., 2016). Specifically, these ten sources are HD-VILA-100M (Xue et al., 2022), InternVid-10M (Wang et al., 2023), VidOR (Shang et al., 2019), VIDAL (YouTube Shorts)(Zhu et al., 2023a), YouCook2(Zhou & Corso, 2017), Charades (Sigurdsson et al., 2016), ActivityNet (Caba Heilbron et al., 2015), Kinetics-700 (Kay et al., 2017), Something-Something v2 (Goyal et al., 2017), and Ego4d (Grauman et al., 2022). These sources offer a wide range of video data from different websites, viewpoints, and domains. The relationship between these ten selected video datasets and others is shown in Fig. 1. The videos from this ten datsets build the video pool for the further video selection. Notably, we use untrimmed videos from each source except for YouCook2 and Kinetics-700. We believe that cutting videos into clips can break the plot continuity, which is essential for understanding the videos.

Based on the video pool, we aim to select dynamic videos. In Figure 1, we outline our criteria for selecting high-quality data. Our main method for identifying dynamic content involves using PySceneDetect, which calculates the number of scenes in a video We found that the number of scenes is a good indicator of video dynamism. Additionally, we have designed a specific approach ④ to exclude videos that mainly contain "slides."

## 3.2 Video Detail Description

**Automated Generation**   For selected videos, we use GPT-4o (OpenAI, 2024) to systematically describe their content. We start by sampling video frames at one frame per second (fps). However, due to the input size constraints of GPT-4o, we cannot use all sampled frames. Instead, we describe the videos sequentially, as shown in Fig 2. We create descriptions at three distinct levels, detailed below.

- *Level-1 Description*: Every 10 seconds, we provide a level-1 description that outlines the events in that segment. This description considers: frames from the current clip and historical context, which includes all recent level-1 descriptions not yet summarized into a level-2 description and the latest level-2 description.
- *Level-2 Description*: Every 30 seconds, we creat a level-2 summary of the entire video plot up to that point. This is based on the last three level-1 descriptions, covering the most recent 30 seconds; and the latest level-2 description.
- *Level-3 Description*: At the video's end, we generate a level-3 description to encapsulate the entire video. The inputs for this description are the recent level-1 descriptions not yet summarized, covering the last moments of the plot after the recent summary; and the latest level-2 description.

## 3.3 Video Question Answering

**Question Type definition**   In addition to detailed video descriptions, our dataset includes a variety of question-answer pairs designed for complex interactions. This setup improves the video understanding model's ability to handle real-life queries. We refer to public video question-answering benchmarks (Xiao et al., 2021;

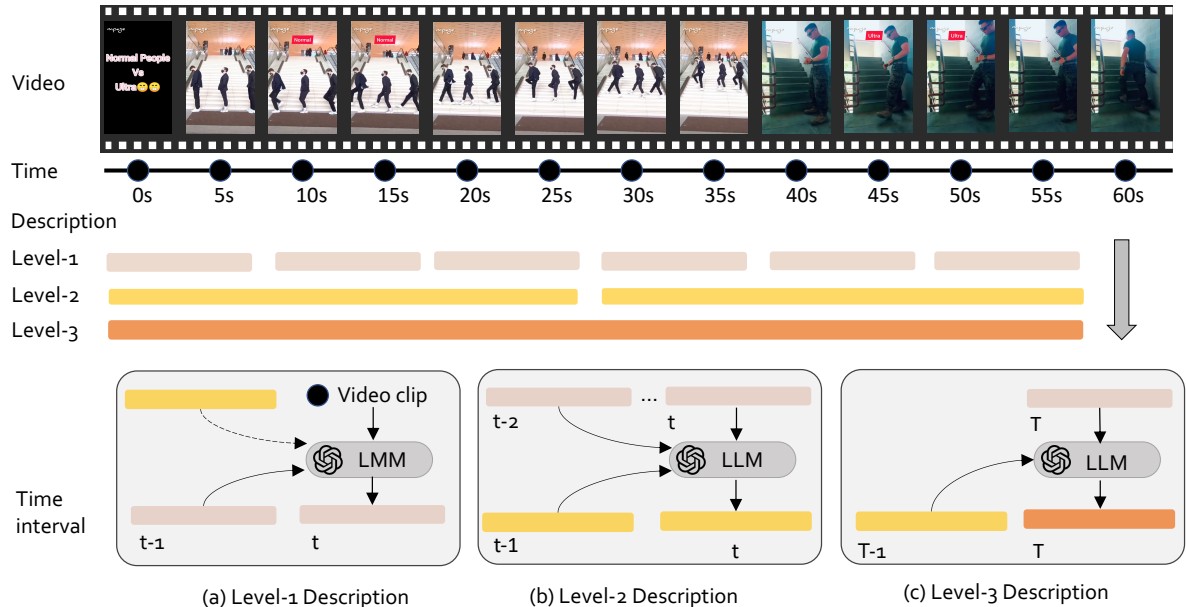

Figure 2: **The video detail description creation pipeline**. A three-level creation pipeline is considered, with each level developed via a recurrent approach. Note that $t$ is the index of time internal at its own level, and $T$ is the last time internal index. (a) To generate the caption for time internal $t$ at level-1, we condition on the current frames in this internal, the caption for time internal $t-1$, and the most recent description summary at level-2 if applicable. (b) To generate caption for time internal $t$ at level-2, we condition on the previous caption at level-2, and captions from three most recent time internals at level-1. (c) To generate the overall caption at the last time internal $T$ at level-3, we condition on the the most recent caption at level-2 and the current caption from level-1.

| Temporal | Q: How do the audiences react after the child hits the pinata correctly? | Spatial | Q: What is behind the 8th man? | Causal | Q: Why do the little boy in red go towards woman in green at first? | Speed | Q: Which is faster, the white car or the bicycle? |
|---|---|---|---|---|---|---|---|
| Binary | Q: Did the child wear shoes while running on the beach? | Count | Q: How many times did the man put his right hand into his pocket? | Plot | Q: How does the interaction between the monkey and the cat indicate? | Description Object | Q: What colors are the railings of the staircase? |
| Time Order | Q: What actions did the person in the red hoodie carry out, and in what order? | Fine-grain Action | Q: Does the person in the video undergo a real physical transformation? | Object Existence | Q: What is the reaction of the audience when the keynote speaker delivers his speech? | Description Human | Q: What does the person on the right's facial expression suggest? |
| Attribute Change | Q: How do the ice cream change? | Camera Direction | Q: Is the camera following the joggers as they move? | Object Direction | Q: Which direction did the man walk towards before exiting the scene relative to the camera? | Description Scene | Q: Where did the rescue operation in the video take place? |

Figure 3: Question types for video question answering in data creation. For each type, we provide its name and an example question.

Yu et al., 2019; khattak et al., 2024; Liu et al., 2024b) to organize these questions into 16 specific categories, as shown in Fig. 3.

**Automated Generation** Given a detailed video description, we use GPT-4o to generate at most one question-answer pair for each type of question. The prompts include: (1) The task definition for the current question type. (2) In-context examples for this type, which include three video descriptions and their three question-answer pairs of this specific type. (3) The detailed video description for the current video. We instruct GPT-4o to return *None* if it cannot generate question-answer pairs for a specific question type.

**Filtering.** To filter out the generated question-answer pairs, we apply the following strategy: (1) remove duplicates using the sentence-transformer (Reimers & Gurevych, 2020), (2) discard answers that begin with

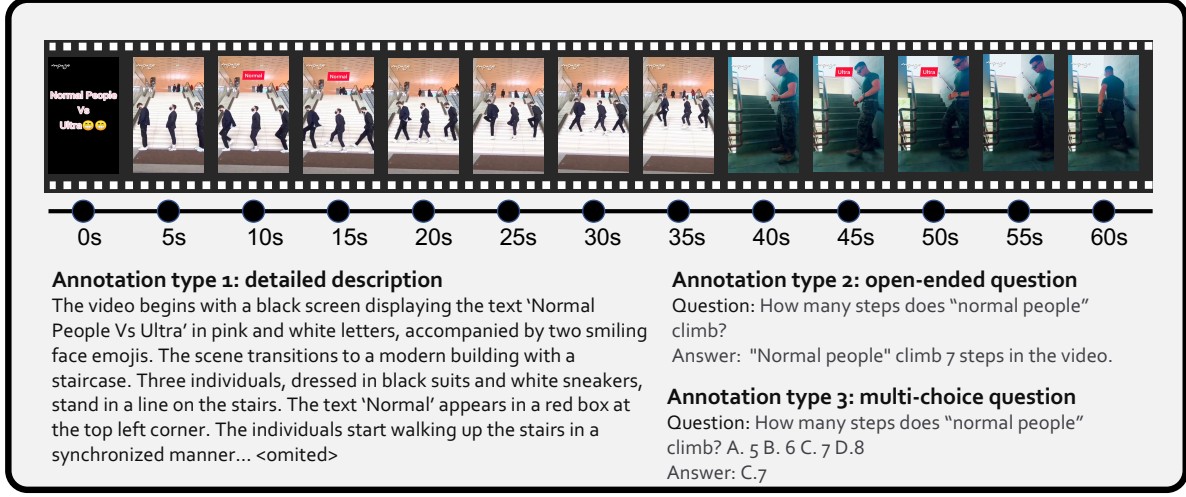

Figure 4: One example to illustrate the video instruction-following data.

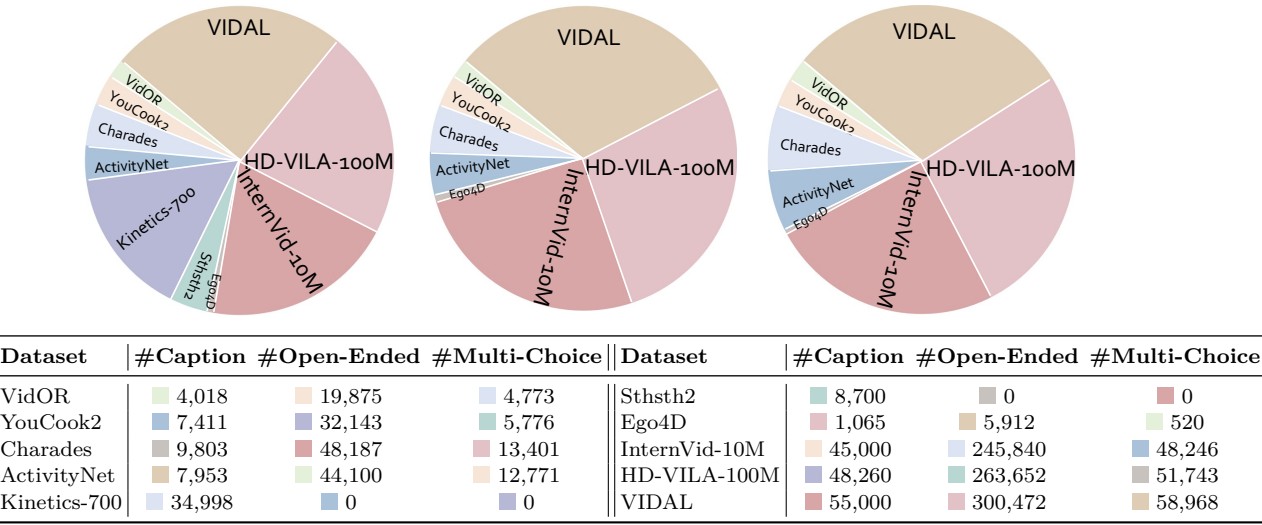

| Dataset | #Caption | #Open-Ended | #Multi-Choice | Dataset | #Caption | #Open-Ended | #Multi-Choice |
|---|---|---|---|---|---|---|---|
| VidOR | 4,018 | 19,875 | 4,773 | Sthsth2 | 8,700 | 0 | 0 |
| YouCook2 | 7,411 | 32,143 | 5,776 | Ego4D | 1,065 | 5,912 | 520 |
| Charades | 9,803 | 48,187 | 13,401 | InternVid-10M | 45,000 | 245,840 | 48,246 |
| ActivityNet | 7,953 | 44,100 | 12,771 | HD-VILA-100M | 48,260 | 263,652 | 51,743 |
| Kinetics-700 | 34,998 | 0 | 0 | VIDAL | 55,000 | 300,472 | 58,968 |

Figure 5: Distribution of data across different datasets and question types (Caption, Open-ended, and Multi-Choice).

phrases like "does not specify," "does not mention," "does not specifically," "does not depict," or "does not show."

## 3.4 Dataset Statistics

**Overview.** We carefully select from our collected data sources to form a balanced and comprehensive collection, resulting in a total of 178K videos and 1.3M instruction-following samples. This includes 178K captions, 960K open-ended QAs, and 196K multiple-choice QAs.

**Dataset Comparison** We provide a comparison of high-quality instruction following video-language datasets, with a focus on synthetic data created with strong AI models, as shown in Table 1. (*i*) *A broad collection of dynamic videos.* In terms of video sources, although LLaVA-Hound (Zhang et al., 2024d) contains the largest number of videos, 44% of its video data are sourced from WebVid (Bain et al., 2021), where most

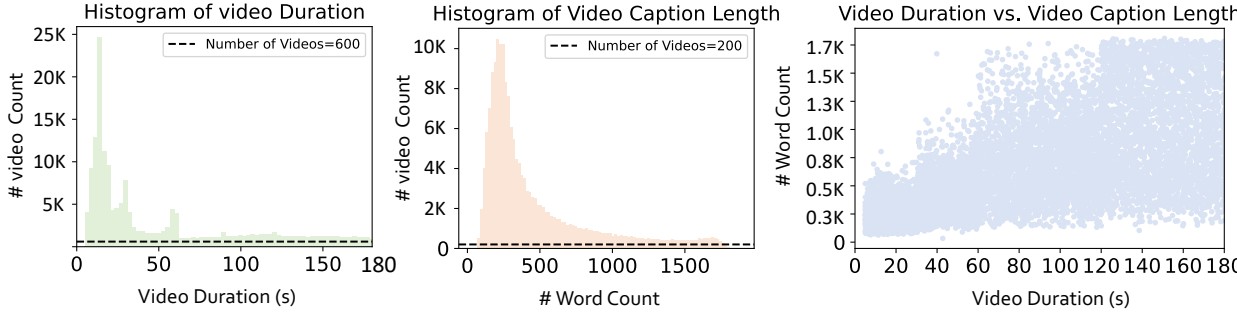

Figure 6: (Left) Visualization of the video duration. (Middle) Visualization of the number of words in the video caption. (Right) Visualization of caption length versus video duration.

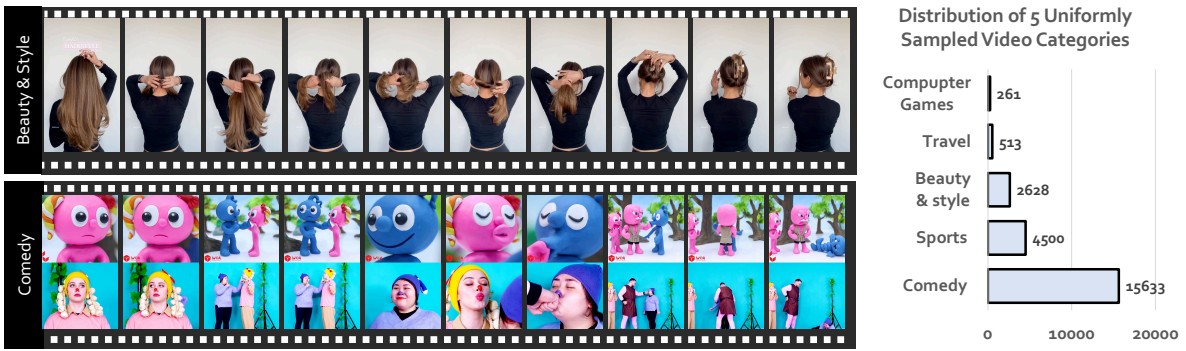

Figure 7: (Left) Display of YouTube Shorts across four video categories. (Right) Distribution of 5 uniformly chosen video categories.

videos are static. ShareGPT4Video (Chen et al., 2024a) includes 30% of its videos from Pexels, Pixabay, and Mixkit, which are aesthetically good but also mostly static. Additionally, the majority of its videos come from Panda-70M, which are short clips from longer videos—suggesting simpler plots. In contrast, we carefully select video sources that offer dynamic, untrimmed videos with complex plots, which are crucial for developing a powerful video understanding model.[1] (*ii*) *High frames per second.* Regarding frame sampling in language annotations, the proposed datasest considers 1 FPS, while other datasets consider much lower FPS. LLaVA-Hound uniformly samples 10 frames from videos of any length. The average FPS is 0.008, which may miss some fine details. ShareGPT4Video picks key frames using CLIP (Radford et al., 2021) based on frame uniqueness. This method might also miss subtle changes in the video because CLIP embeddings do not capture fine-grained dynamics well. Our method samples FPS=1 without using key frame selection algorithms, ensuring the detailed temproal information can be expressed in annotations and high coverage. (*iii*) *Diverse tasks.* The proposed dataset considers three common task types, including caption, free-form and closed-form QA, while existing datasets only consider a subset. Meanwhile, the quality and numbers of samples in our dataset is higher.

## 4 Experiments

We conducted evaluations for the LLaVA-Video models across all benchmarks using LMMs-Eval (Zhang et al., 2024a) to ensure standardization and reproducibility. To fairly compare with other leading video LMMs, we primarily used results from original papers. When results were not available, we integrated the models into LMMs-Eval and assessed them under consistent settings. Following LLaVA-OneVision (Li et al., 2024c), we employed SigLIP (Zhai et al., 2023) as our vision encoder, and Qwen2 (Yang et al., 2024) as the LLM. The

---

[1]Example videos: WebVid, Pixabay, Pexels, Mixkit.

Table 1: **Comparison of LLaVA-Video-178K and other video-language datasets**. Average FPS represents the average number of frames per second that are used to prompt GPT-4o/GPT-4V for annotation. ★ VIDAL, WebVid, ActivityNet. ■ Panda-70M, Pexels, Pixabay, Mixkit, BDD100K, Ego4d. ✪ HD-VILA-100M, Kinetics-700M, Ego4D, VidOR, InternVid, YouCook2, ActivityNet, Sth-sthv2, VIDAL, Charades.

| | Text | Video Source | #Video | Total Video Length | Average FPS | #Caption | #OE QA | #MC QA |
|---|---|---|---|---|---|---|---|---|
| LLaVA-Hound | GPT-4V | ★ | 900K | 3Khr | 0.008 | 900K | 900K | 0 |
| ShareGPT4Video | GPT-4V | ■ | 40K | 0.2Khr | 0.15 | 40K | 0 | 0 |
| LLaVA-Video-178K | GPT-4o | ✪ | 178K | 2Khr | 1 | 178K | 960K | 196K |

LLaVA-Video model builds on the single-image (SI) stage checkpoint from the LLaVA-OneVision model (Li et al., 2024c), which was trained using only image data.

**Video Representations** Following the classic SlowFast idea in video representations (Feichtenhofer et al., 2019; Xu et al., 2024b; Huang et al., 2024), we develop *LLaVA-Video $_{SlowFast}$* to optimize the balance between the number of frames and the count of visual tokens, within the budget of the limited context window in LLM and GPU memory for video representation. Please refer to Appendix 7 for detailed information. Specifically, we represent each video as a sequence with maximum $T$ frames. Each frame is represented in $M$ tokens. we categorize the frames into two groups, based on the a strike rate $s$, where the every $s$ frames are uniformly selected to form the *slow* frame group, and the rest of the frames are consdiered as the *fast* frame group. Note that a special case $s = 1$ leads to only one group, reducing the SlowFast representation to the original simple representation. For each group, we apply different pooling rate using Pytorch function pooling `avg_pool2d()`. $p \times p$ pooling and $2p \times 2p$ pooling for slow and fast frames, respectively. To summarize, we paramterize the video representation configuration as $\mathcal{V} = (T, M, s, p)$. The total number of tokens is $\#tokens = \lfloor T/s \rfloor \times \lfloor M/p^2 \rfloor + (T - \lfloor T/s \rfloor) \times \lfloor M/4p^2 \rfloor$

**Evaluation Benchmarks.** For full evaluation, we consdier 11 video benchmarks. conducted tests across various video captioning , video open-ended question-answering and video multiple-choice question-answering benchmarks, including ActivityNet-QA (Yu et al., 2019), which features human-annotated action-related QA pairs from the ActivityNet dataset. We also utilized LongVideoBench (Wu et al., 2024b), EgoSchema (Mangalam et al., 2024), and MLVU (Zhou et al., 2024) for long video understanding, PerceptionTest (Pătrăucean et al., 2023) for assessing fine-grained perception skills, and VideoMME (Fu et al., 2024) and NExT-QA (Xiao et al., 2021) for diverse video domains and durations. Additional tests included VideoDetailCaption (LMMs-Lab, 2024), Dream-1K (Wang et al., 2024), Video-ChatGPT (Maaz et al., 2024) for detailed video descriptions, TemporalBench Cai et al. (2024) for fine-grained temporal understanding.

For ablation studies in . 4.2 and Sec. 4.3, we conduct evaluation across 4 datasets. NExT-QA (Xiao et al., 2021) and PerceptionTest (Pătrăucean et al., 2023), which use training data from the LLaVA-Video-178K, are treated as in-domain datasets. Conversely, VideoMME (Fu et al., 2024) and EgoSchema (Mangalam et al., 2024) are consider as zero-shot datasets.

## 4.1 Overall Results

We fine-tune LLaVA-OneVision (SI) on the joint dataset of video and image data. Specifically, we added video data from the LLaVA-Video-178K dataset and four public datasets: ActivityNet-QA (Yu et al., 2019), NExT-QA (Xiao et al., 2021), PerceptionTest (Pătrăucean et al., 2023), and LLaVA-Hound-255K (Zhang et al., 2024d), focusing on videos shorter than three minutes. These datasets were selected to improve our model's performance, contributing to a total of 1.6 million video-language samples, which include 193,510 video descriptions, 1,241,412 open-ended questions, and 215,625 multiple-choice questions. Remarkably, 92.2% of the video descriptions, 77.4% of the open-ended questions, and 90.9% of the multiple-choice questions were newly annotated. Additionally, we used 1.1 million image-language pairs from the LLaVA-OneVision model (Li et al., 2024c). We consider the same video representation configurations for the training and inference

Table 2: LLaVA-Video performance on video benchmarks. We report the score out of 5 for VideoDC, VideoChatGPT while other results are reported in accuracy. All results are reported as 0-shot accuracy. *indicates that the training set has been observed in our data mixture.

| | Caption | | Open-Ended Q&A | | Multi-Choice Q&A | | | | | | | |
| Model | VideoDC | Dream-1K | ActNet-QA | VideoChatGPT | EgoSchema | MLVU | MVBench | NExT-QA | PerceptionTest | LongVideoBench | TemporalBench | VideoMME |
| | test | test | test | test | test | m-avg | test | mc | val | val | m-acc | wo/w-subs |
| *Proprietary models* | | | | | | | | | | | | |
| GPT-4o (OpenAI, 2024) | - | 39.2 | - | - | - | 64.6 | - | - | - | 66.7 | 35.3 | 71.9/77.2 |
| Gemini-1.5-Pro (Team et al., 2023) | - | 36.2 | 57.5 | - | 72.2 | - | - | - | - | 64.0 | 25.6 | 75.0/81.3 |
| *Open-source models* | | | | | | | | | | | | |
| VILA-40B (Lin et al., 2024) | 3.37 | 33.2 | 58.0 | 3.36 | 58.0 | - | - | 67.9 | 54.0 | - | - | 60.1/61.1 |
| PLLaVA-34B (Xu et al., 2024a) | - | 28.2 | 60.9 | 3.48 | - | - | 58.1 | - | - | 53.2 | - | - |
| LongVA-7B (Zhang et al., 2024c) | 3.14 | - | 50.0 | 3.20 | - | 56.3 | - | 68.3 | - | - | - | 52.6/54.3 |
| IXC-2.5-7B (Zhang et al., 2024b) | - | - | 52.8 | 3.46 | - | 37.3 | 69.1 | 71.0 | 34.4 | - | 16.7 | 55.8/58.8 |
| LLaVA-OV-7B (Li et al., 2024c) | 3.75 | 31.7 | 56.6 | 3.51 | 60.1 | 64.7 | 56.7 | 79.4* | 57.1 | 56.5 | 18.7 | 58.2/61.5 |
| VideoLLaMA2-72B (Cheng et al., 2024) | - | 27.1 | 55.2 | 3.16 | 63.9 | 61.2 | 62.0 | - | - | - | - | 61.4/63.1 |
| LLaVA-OV-72B (Li et al., 2024c) | 3.60 | 33.2 | 62.3 | 3.62 | 62.0 | 68.0 | 59.4 | 80.2* | 66.9 | 61.3 | 26.6 | 66.2/69.5 |
| LLaVA-Video-7B | 3.66 | 32.5 | 56.5* | 3.52 | 57.3 | 70.8 | 58.6 | 83.2* | 67.9* | 58.2 | 22.9 | 63.3/69.7 |
| LLaVA-Video-72B | 3.73 | 34.0 | 63.4* | 3.62 | 65.6 | 74.4 | 64.1 | 85.4* | 74.3* | 61.9 | 33.7 | 70.5/76.9 |

stages. On 128 NVIDIA H100 GPUs, the video representations for LLaVA-Video-7B and LLaVA-Video-72B are $\mathcal{V} = (64, 679, 1, 2)$ and $\mathcal{V} = (64, 679, 3, 2)$, respectively.

In Table 2, we compare the performance of different models on various video benchmarks. The 72B model performs as well as the commercial, closed-source model Gemini-1.5-Flash (Team et al., 2023), highlighting the effectiveness of open-source efforts in achieving comparable results. The LLaVA-Video-7B model outperforms the previous top model, LLaVA-OV-7B, in seven out of ten datasets. Analysis of individual datasets shows some noteworthy trends. For instance, on benchmarks like MLVU, LongVideoBench, and VideoMME, which primarily use video data from YouTube, this improvement may be due to the inclusion of extensive YouTube data in LLaVA-Video-178K, as illustrated in Fig. 5. Additionally, the improvement on ActivityNet-QA is small; this could be because many questions in ActivityNet-QA, such as "What's the color of the ball?" can be answered by viewing a single frame. The visibility of the ball from the beginning to the end of the video means understanding the video sequence is unnecessary, so LLaVA-Video-178K offers little advantage in this context. We find that LLaVA-Video-7B is notably weaker in the specialized task of EgoSchema, an ego-centric dataset. This weakness may be due to a significant reduction in the proportion of ego-centric data in the training dataset of LLaVA-Video. However, this impact is less pronounced in larger models, as demonstrated by the LLaVA-Video-72B model's superior performance over LLaVA-OV-72B in EgoSchema.

### 4.2 Dataset Ablation

Note that the training set for LLaVA-Video includes six datasets: LLaVA-Video-178K, LLaVA-Hound (Zhang et al., 2024d), NExT-QA (Xiao et al., 2021), ActivityNet-QA (Yu et al., 2019), PerceptionTest (Pătrăucean et al., 2023), and image data from LLaVA-OneVision (Li et al., 2024c). In this section, we conduct ablation studies to assess the impact of each dataset. We separately fine-tune the LLaVA-OneVision (SI) model for each experimental setting, progressively adding datasets to the baseline.

The results are presented in Table 3. Initially, we used a basic model trained solely on the LLaVA-Hound dataset as our baseline. Compared to this baseline, adding the LLaVA-Video-178K dataset significantly

Table 3: Ablation study on the LLaVA-Video model with various configurations of training data. Three Q&A datasets indicate: NExT-QA, ActivityNet-QA and PerceptionTest.

| Method | in-domain | | out-of-domain | |
|---|---|---|---|---|
| | NExT-QA | PercepTest | EgoSchema | VideoMME |
| | mc | val | test | wo |
| LLaVA-Hound | 64.4 | 51.4 | 51.0 | 54.1 |
| +LLaVA-Video-178K | 80.1 | 57.1 | 56.5 | 63.2 |
| +Three Q&A datasets | 80.1 | 69.0 | 55.6 | 61.9 |
| +LLaVA-OV (images) | 83.2 | 67.9 | 57.3 | 63.4 |

Table 4: Comparison of LLaVA-Video-178K and other video instruction-following datasets.

| | #Cap | #OE | #MC | in-domain | | out-of-domain | |
|---|---|---|---|---|---|---|---|
| | | | | NExT-QA | PercepTest | EgoSchema | VideoMME |
| | | | | mc | val | test | wo |
| LLaVA-Hound | 900K | 900k | 0 | 39.8 | 53.1 | 25.8 | 55.2 |
| LLaVA-V-178K | 178K | 900k | 0 | 73.2 | 55.9 | 49.8 | 59.6 |
| ShareGPT4Video | 40K | 40K | 19K | 69.6 | 55.2 | 58.9 | 51.0 |
| LLaVA-V-178K | 40K | 40K | 19K | 75.8 | 55.4 | 55.8 | 53.5 |

improved performance, enhancing scores in both in-domain and out-of-domain tasks. Specifically, we observed a 31.9-point increase in NExT-QA scores and a 9.1-point rise in VideoMME scores. Furthermore, including the PerceptionTest dataset enhanced its associated task. Additionally, integrating high-quality image data provided modest benefits on EgoSchema.

## 4.3 Dataset Comparison

We conduct two ablation studies to analyze our dataset and training strategy. In Table 4, we compared three datasets where the language annotations are from GPT-4V/GPT-4o. For each experiment, we fine-tune the LLaVA-OneVision (SI) model separately on each specific dataset setting.

Two group of experiments are considered to assess the data quality of LLaVA-Video-178K compare to LLaVA-Hound and ShareGPT4Video. In the first group, to compare LLaVA-Video-178K with LLaVA-Hound, we randomly selected 900K open-ended questions to match the number in LLaVA-Hound. We included all captions and did not sample the multiple-choice questions. In the second group, comparing LLaVA-Video-178K to ShareGPT4Video, we randomly sampled 40K video captions to align with those in ShareGPT4Video. Since ShareGPT4Video lacks open-ended and multiple-choice questions, we supplemented with annotations from NExT-QA, PerceptionTest, and ActivityNet-QA. In the first group of Table 4, we compare LLaVA-Video-178K with LLaVA-Hound. Although LLaVA-Hound has more captions than LLaVA-Video-178K, our results are still better. The quality of LLaVA-Hound is limited due to two main issues: (1) Static video: Its primary video source is WebVid (Bain et al., 2021), which tends to have relatively static content. (2) Sparse sampling: its sampling rate of 10 frames per video leads to annotations that do not fully capture the complete plot of the video. This underscores that the quality of video instruction-following data is more important than its quantity. Additionally, the second experiment group in Table 4 shows that the model trained with LLaVA-Video-178K outperforms that of ShareGPT4Video, highlighting the superiority of our data's quality.

## 5   Conclusion

This study introduces the LLaVA-Video-178K dataset, a high-quality synthetic dataset for video-language instruction-following. It is favored for its dense frame sampling rate in longer, untrimmed videos, covering diverse tasks such as captioning, open-ended and multi-choice QA. By training on the joint dataset of LLaVA-Video-178K with existing visual instruction tuning data, we developed a new model family, LLaVA-Video, which considers video representation to effectively use GPU resources. This allows us to include more frames in the training process. The experimental results have demonstrated the effectiveness of the proposed synthetic dataset, and LLaVA-Video models have achieved excellent performance on a wide range of video benchmarks.

## 6   Limitations

The videos in LLaVA-Video-178K are sourced from various platforms. This diversity introduces potential biases inherent in these sources. Furthermore, there is a concern regarding the potential skew in the question-answer pairs, possibly influenced by the annotators' perspectives.

## Acknowledgement

This study is supported by the Ministry of Education, Singapore, under its MOE AcRF Tier 2 (MOE-T2EP20221-0012, MOE-T2EP20223-0002), and under the RIE2020 Industry Alignment Fund – Industry Collaboration Projects (IAF-ICP) Funding Initiative, as well as cash and in-kind contribution from the industry partner(s).

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

# 7 Video Representations

## 7.1 Efficient Video Representations in LMMs

Current designs of large multimodal models (LMM) typically connect a vision encoder (Radford et al., 2021; Zhai et al., 2023) to a large language model (Yang et al., 2024) through a lightweight projector (Liu et al., 2024a) or a resampler (Li et al., 2023; Alayrac et al., 2022). These components transform a set of visual representations into "visual tokens" aligned with text embeddings. In contrast to image-based LMMs, which generate only a small number of visual tokens easily managed by a standard GPU, video LMMs face challenges due to a large number of visual tokens derived from multiple video frames. The LLaVA-NeXT-Video (Zhang et al., 2024e) and PLLaVA (Xu et al., 2024a) models address this by simly considering average pooling to reduce the number of tokens representing each frame.

Following the idea of SlowFast in the traditional video understanding (Feichtenhofer et al., 2019), adaptive reductions in visual tokens are demonstrated by recent video LMMs, LITA (Huang et al., 2024) and SlowFast-LLaVA (Xu et al., 2024c). Initially, these methods represent all sampled frames with a minimal number of visual tokens (fast frame)— typically just one—by using a large pooling stride. They then switch to a smaller pooling stride for certain frames to retain more visual tokens (slow frame). Finally, they combine the visual tokens of fast frames with those of slow frames. However, this approach can lead to some frames being represented twice. In contrast, our method uses a larger pooling stride for sampled frames to maintain fewer visual tokens (fast frame) *or* a smaller stride for others to keep more (slow frame). We then arrange slow and fast frames in an interleaving pattern.

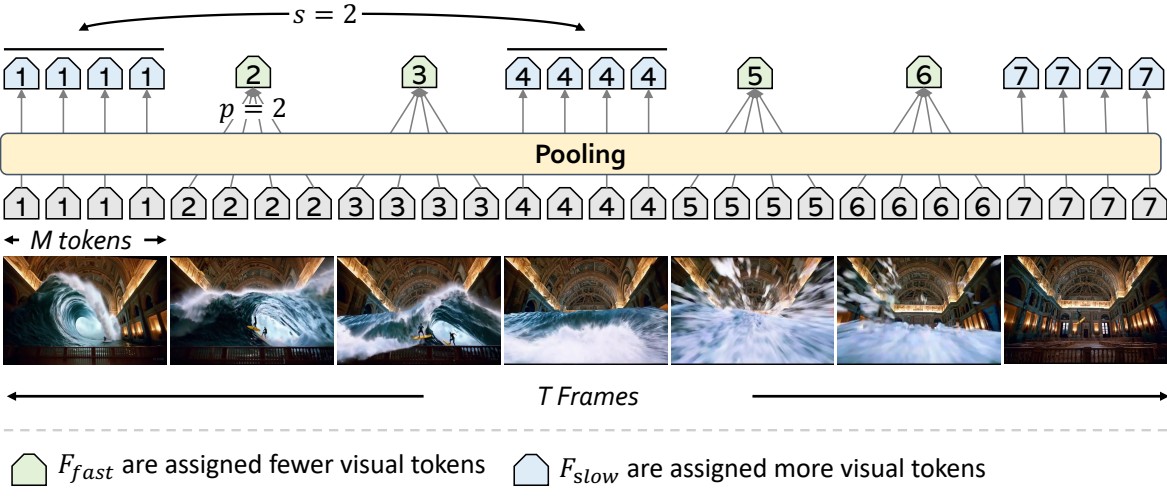

Figure 8: Video representations. A different number of tokens are utilized to represent frames.

## 7.2 LLaVA-Video SlowFast

We represent each video as a sequence with maximum $T$ frames. Each frame is represented in $M$ tokens. FPS-based video representation can be considered in the future. Specifically, each frame is encoded via an image encoder and a two-layer MLP for projection. These visual tokens are concatenated with word tokens and processed by a large language model (LLM). Managing tokens for every frame can be computationally demanding. For instance, employing the SigLIP (Zhai et al., 2023) encoder for a video with $T = 100$ results in 67,600 tokens, assuming $M = 729$ tokens per frame, which often exceeds GPU memory limits. This issue is exacerbated when using large-parameter LLMs; with the Qwen2-72B model, we could only process 8 frames before maxing out the memory on 128 NVIDIA H100 GPUs. Such a limited number of frames can introduce inconsistencies in language annotations, reducing model efficacy. One strategy to incorporate more frames is by applying $p \times p$ spatial average pooling to reduce $M$ to $M/p^2$, thus lowering the token count per frame as

suggested by recent studies (Xu et al., 2024a; Zhang et al., 2024e). However, the number of visual tokens is crucial for preserving the informational content of each frame, which is vital for video comprehension.

In our LLaVA-Video $_{\texttt{SlowFast}}$, we categorize the frames into two groups, based on the a strike rate $s$, where the every $s$ frames are uniformly selected to form the *slow* frame group, and the rest of the frames are consdiered as the *fast* frame group. Note that a special case $s = 1$ leads to only one group, reducing the SlowFast representation to the original simple representation. For each group, we apply different pooling rate using Pytorch function pooling `avg_pool2d()`. $p \times p$ pooling and $2p \times 2p$ pooling for slow and fast frames, respectively. To summarize, we paramterize the video representation configuration as $\mathcal{V} = (T, M, s, p)$. The total number of tokens is $\#tokens = \lfloor T/s \rfloor \times \lfloor M/p^2 \rfloor + (T - \lfloor T/s \rfloor) \times \lfloor M/4p^2 \rfloor$

# 8 Data

## 8.1 Video Detail Description

As discussed in Section 3.2, we show that generating *level-1 description* should consider historical context. Figure 9 illustrates the impact of excluding historical context on the quality of video descriptions. Specifically, including historical context helps accurately identify characters across different times as the same individual.

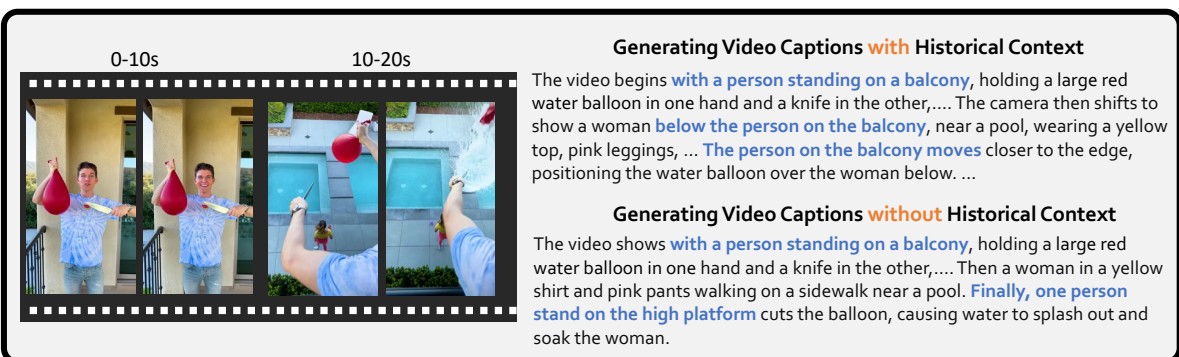

Figure 9: Generating video captions with or without historical context.

## 8.2 Video Question Answering

In Table 5, we list the names and descriptions of different question types and their corresponding proportions in the LLaVA-Video-178K dataset. The prompt used to generate video question-answer pairs from GPT-4O is shown in Table. 6. In Fig. 4, we show an example of a video along with its detailed description, an open-ended question, and a multiple-choice question.

## 8.3 Dataset Comparison

We provide a more comprehensive comparison of LLaVA-Video-178K with other video-language datasets for the video caption task and video question answer task. Specifically, we organize the table into four groups, each characterized by its method of text annotation. As shown in Table 7, unlike other datasets, LLaVA-Video-178K uniquely includes all three types of annotations: captions, open-ended questions, and multiple-choice questions.

# 9 Beyond Singularity: Extensive Sampling Matters

We perform experiments to explore how video representations affect the model's performance. All experiments were carried out in a video-only setting, using video data with durations from 0 to 30 seconds as our training data. We focused on evaluating how the number of frames and the number of visual tokens per frame impact

Table 5: Question types for video question answering in data creation. For each type, we provide its name, description, and the proportion it represents in the LLaVA-Video-178K.

| Question type | Description | Proportion |
|---|---|---|
| Temporal | Designed to assess reasoning about temporal relationships between actions/events. Questions involve previous, present, or next actions. | 7.2% |
| Spatial | Tests ability to perceive spatial relationships between observed instances in a video scene. | 7.2% |
| Causal | Focuses on explaining actions/events, determining intentions of actions or causes for subsequent events. | 7.2% |
| Description-Scene | Assesses ability to describe the major scene of the video, like where it takes place and the overall environment. | 7.2% |
| Description-Human | Involves describing actions or attributes of people, such as their activities and appearances. | 6.7% |
| Description-Object | Assesses ability to describe attributes of objects, like their appearance and function. | 7.0% |
| Count | Tests ability to count instances of objects, people, actions, and to distinguish between old and new elements in a scene. | 7.1% |
| Binary | Involves yes or no questions related to the video content. | 7.2% |
| Fine Grained Action Understanding | Creates questions challenging comprehension of subtle actions. | 6.5% |
| Plot Understanding | Challenges ability to interpret the plot in the video. | 7.1% |
| Non-Existent Actions with Existent Scene Depictions | Assesses reasoning with introduced non-exist ent activities without changing physical details. | 6.6% |
| Time Order Understanding | Challenges recognition of temporal sequence of activities in videos. | 6.9% |
| Object Direction | Emphasizes perception of object movement direction. | 3.8% |
| Camera Direction | Focuses on the direction of camera movement. | 4.1% |
| Speed | Delves into discerning variations in speed, including absolute and relative speeds. | 3.6% |
| Attribute Change | Centers on how attributes of objects or the entire video change over time, like size, shape, color, and more. | 4.5% |

model performance. Regarding the frame count, it is noteworthy that observing the effects of a high number of frames—such as over 100—does not necessarily require long videos. Our results indicate that the dynamic properties of the data render even 100 frames insufficient to fully capture the condent of a 30-second video, which typically runs at 15 FPS.

In Table 8, the first group shows an increase in the number of frames from 32 to 110. We set 110 frames as the upper limit to avoid overloading the GPU. With more frames, we see significant improvements in all datasets. While it's generally expected that using more frames boosts performance, previous studies (Luo et al., 2021; Lei et al., 2021; 2022) have noted that performance tends to plateau when training with more than 16 frames. We propose that the saturation observed in earlier studies arises due to the selection of training datasets such as MSVD (Chen & Dolan, 2011) and WebVid (Bain et al., 2021), where the video content is highly static, allowing a small number of frames to represent the entire video effectively. In contrast, the dynamic nature of the videos and the detailed nature of the annotations in LLaVA-Video-178K allow for continuous benefits from extensive sampling

The second group in Table 8 demonstrates the effects of varying the number of inference frames while keeping the number of training frames constant. A modest increase in the inference frames slightly enhances performance; however, excessively increasing the number of inference frames can degrade it.

```
tasks = "
# Temporal: this task is designed to assess the capability of reasoning ...<omitted>
## caption-1: The video features a child sitting in a baby chair at a dining table, creating...<omitted>
## question-1: What was the child doing as he sat on the baby chair?
## answer-1: The child was reading a book.
...
## caption-3: ...<omitted>
## question-3: ...<omitted>
## answer-3: ...<omitted>
# Spatial: this task involves creating questions that test a person's ability...<omitted>
...<omitted> "
system_message = "
### Task:
Given a detailed description that summarizes the content of a video, generate question-answer pairs based
on the description to help humans better understand the video. The question-answer pairs should be faithful
to the content of the video description and developed from different dimensions to promote comprehensive
understanding of the video.
Here are some question dimensions and their explanations and exampled question-answer pairs for reference:
{task_definitions}
#### Guidelines For Question-Answer Pairs Generation:
- Read the video description provided carefully, paying attention to the content, such as the scene where the
video takes place, the main characters and their behaviors, and the development of the events.
- Generate appropriate question-answer pairs based on the description. The question-answer pairs should
cover as many question dimensions and not deviate from the content of the video description.
- Generate 1 question-answer pair for each dimension.
### Output Format:
1. Your output should be formed in a JSON file.
2. Only provide the Python dictionary string.
Your response should look like:
["Dimension":  <dimension-1>, "Question":  <question-1>, "Answer":  <answer-1>,
"Dimension":  <dimension-2>, "Question":  <question-2>, "Answer":  <answer-2>...] "
user_message = "
Please generate question-answer pairs for the following video description:
Description: {caption} "

for cur_video in videos:
    sys_msg = system_messages.format(task_definitions=tasks)
    usr_msg = user_messages.format(caption=cur_video)
    response = GPT4O(sys_msg,usr_msg)
```

Table 6: We explain the process of creating prompts for GPT-4O to gather question-answer pairs from each video description. `tasks` includes the definition of all question types along with examples of question-answer pairs. We instruct GPT-4O to generate questions that cover as many question types as possible.

In Table 8's third group, we illustrates the trade-off between the number of frames and the number of tokens per frame. Configurations with fewer tokens per frame but more frames yield superior results, even with a lower total count of visual tokens (18,590 versus 21,632). This finding emphasizes that increasing the number of frames, rather than the tokens per frame or the total number of tokens, enhances performance. However, a balance is necessary; as the number of frames increases to 440 and the tokens per frame decreases to 64, performance drops. This observation led us to use LLaVA-Video $_{\text{SlowFast}}$ for video representation.

Table 7: **Comparison of LLaVA-Video-178K and other video-language datasets**. Average FPS represents the average number of frames per second that are used to prompt GPT-4o/GPT-4V for annotation.

| | Text | #Video | Total Video Length | Average FPS | #Caption | #OE QA | #MC QA |
|---|---|---|---|---|---|---|---|
| HowTo100M (Miech et al., 2019) | ASR | 136M | 134.5Khr | - | 136M | 0 | 0 |
| ACAV (Lee et al., 2021) | ASR | 100M | 277.7Khr | - | 100M | 0 | 0 |
| YT-Temporal-180M (Zellers et al., 2021) | ASR | 180M | - | - | 180M | 0 | 0 |
| HD-VILA-100M (Xue et al., 2022) | ASR | 103M | 371.5Khr | - | 103M | 0 | 0 |
| MSVD (Chen & Dolan, 2011) | Manual | 1970 | 5.3h | - | 1K | 0 | 0 |
| LSMDC (Rohrbach et al., 2015) | Manual | 118K | 158h | - | 118K | 0 | 0 |
| MSR-VTT (Xu et al., 2016) | Manual | 10K | 40h | - | 10K | 0 | 0 |
| DiDeMo (Anne Hendricks et al., 2017b) | Manual | 27K | 87h | - | 27K | 0 | 0 |
| ActivityNet (Caba Heilbron et al., 2015) | Manual | 100K | 849h | - | 100K | 0 | 0 |
| YouCook2 (Zhou & Corso, 2017) | Manual | 14K | 176h | - | 14K | 0 | 0 |
| TVQA (Lei et al., 2018) | Manual | 21K | 3.39Khr | - | 0 | 0 | 152K |
| ActivityNet-QA (Yu et al., 2019) | Manual | 5.8K | 290h | - | 0 | 58K | 0 |
| Social-IQ (Zadeh et al., 2019) | Manual | 1.2K | 20h | - | 0 | 0 | 7.5k |
| NExT-QA (Xiao et al., 2021) | Manual | 5.4K | 66h | - | 0 | 52K | 47K |
| MSVD-QA (Xu et al., 2017) | Open-source Model | 1.9K | 5.3h | - | 41K | 50K | 0 |
| MSRVTT-QA (Xu et al., 2017) | Open-source Model | 10K | 40h | - | 0 | 243K | 0 |
| Panda-70M (Chen et al., 2024b) | Open-source Model | 70.8M | 166.8Khr | - | 70.8M | 0 | 0 |
| LLaVA-Hound (Zhang et al., 2024d) | GPT-4V | 900K | 3Khr | 0.008 | 900K | 900K | 0 |
| ShareGPT4Video (Chen et al., 2024a) | GPT-4V | 40K | 0.2Khr | 0.15 | 40K | 0 | 0 |
| LLaVA-Video-178K | GPT-4o | 178K | 2Khr | 1 | 178K | 960K | 196K |

Table 8: Visual Representation Configurations and Performance Correlation. $T^{\text{train}}$ and $T^{\text{test}}$ are the number of frames in the training and inference stage, respectively. $M/p^2$: number of visual tokens per frame.

| | | | in-domain | | out-of-domain | |
|---|---|---|---|---|---|---|
| | | | **NExT-QA** | **PerceptionTest** | **EgoSchema** | **VideoMME** |
| $T^{\text{train}}$ | $T^{\text{test}}$ | $M/p^2$ | mc | val | test | wo |
| *Training with more frames* | | | | | | |
| 32 | 32 | 169 | 80.4 | 68.2 | 56.3 | 59.1 |
| 64 | 64 | 169 | 81.4 (+1.0) | 68.3 (+0.1) | 58.4 (+2.1) | 59.6 (+0.5) |
| 110 | 110 | 169 | 82.0 (+1.6) | 68.3 (+0.1) | 59.1 (+2.8) | 60.4 (+1.3) |
| *Inference with more frames* | | | | | | |
| 32 | 32 | 169 | 80.4 | 68.2 | 56.3 | 59.1 |
| 32 | 64 | 169 | 80.7 (+0.3) | 68.9 (+0.7) | 56.3 (+0.0) | 59.9 (+0.8) |
| 32 | 110 | 169 | 80.5 (+0.1) | 67.2 (-1.0) | 55.2 (-1.1) | 58.8 (-0.3) |
| *Using more frames with fewer visual tokens per frame* | | | | | | |
| 32 | 32 | 729 | 79.4 | 69.5 | 58.3 | 59.1 |
| 110 | 110 | 169 | 82.0 (+2.6) | 68.3 (-1.2) | 59.1 (+0.8) | 60.4 (+1.3) |
| 440 | 440 | 64 | 81.6 (+2.2) | 67.2 (-2.3) | 59.4 (+1.1) | 60.2 (+1.1) |

# 10 Capabilities

Beyong achieve good benchmark performance, Our observations of LLaVA-Video reveal various capabilities in video understanding. Specifically, it show a great abilities in the understanding video using real-world knowledge,including, but not limited to:

Table 9: Comparison of different video representations. The video representation $\mathcal{V}$ is consistent in training and inference for all methods, except that SlowFast-LLaVA considers simple representation $\mathcal{V}$ in training and its specified $\mathcal{V}$ in inference.

| Method | $\mathcal{V} = (T, M, s, p)$ | #Visual Tokens | in-domain | | out-of-domain | |
| | | | NExT-QA | PerceptionTest | EgoSchema | VideoMME |
| | | | mc | val | test | wo |
|---|---|---|---|---|---|---|
| Simple representation | (32, 729, 1, 2) | 5,408 | 80.4 | 68.2 | 56.3 | 59.1 |
| LLaVA-Video $_{\text{SlowFast}}$ | (64, 729, 3, 2) | 5,396 | 81.1 | 67.7 | 57.1 | 59.8 |
| LITA | (42, 729, 2, 2) | 5,313 | 80.8 | 68.3 | 54.3 | 59.1 |
| SlowFast-LLaVA | (42, 729, 2, 2) | 5,313 | 79.4 | 68.2 | 56.2 | 58.9 |

- *Optical Illusion*: As shown in Table 11, LLaVA-Video recognizes that the green dragon in the video is not a real 3D object. It appears three-dimensional due to an optical illusion that affects human perception.
- *Special Domain*: As indicated in Table 11, LLaVA-Video understands the content within special domains in the video, such as sketches and fights in video games.
- *Unusual Action*: As detailed in Table 12, LLaVA-Video identifies atypical actions in the video, such as "physical therapy" for pets, beyond ordinary activities.
- *Physical Laws*: As shown in Table 13, LLaVA-Video comprehends basic physical laws demonstrated in the video, like zero gravity in space stations, which allows objects to float without falling.

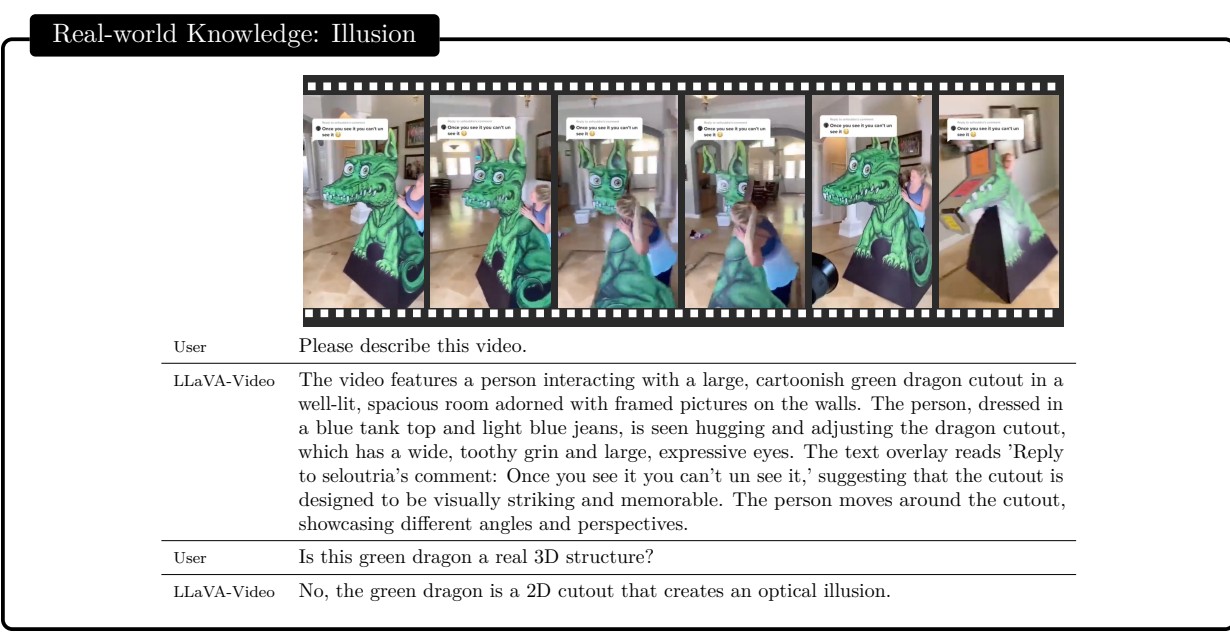

**Real-world Knowledge: Illusion**

| | |
|---|---|
| User | Please describe this video. |
| LLaVA-Video | The video features a person interacting with a large, cartoonish green dragon cutout in a well-lit, spacious room adorned with framed pictures on the walls. The person, dressed in a blue tank top and light blue jeans, is seen hugging and adjusting the dragon cutout, which has a wide, toothy grin and large, expressive eyes. The text overlay reads 'Reply to seloutria's comment: Once you see it you can't un see it,' suggesting that the cutout is designed to be visually striking and memorable. The person moves around the cutout, showcasing different angles and perspectives. |
| User | Is this green dragon a real 3D structure? |
| LLaVA-Video | No, the green dragon is a 2D cutout that creates an optical illusion. |

Table 10: LLaVA-Video learns to understand the *optical illusion* in the video.

**Real-world Knowledge: Special Domain**

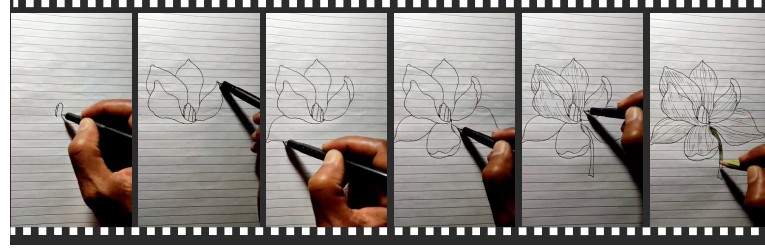

| User | Please describe this video. |
|---|---|
| LLaVA-Video | The video begins with a blank, lined sheet of paper. A hand holding a black pen starts drawing on the paper, creating the outline of a flower with five petals. The artist adds details to the petals, including lines and shading to give them a more realistic appearance. The stem and leaves are also drawn, completing the flower illustration. The artist adds final touches to the flower drawing, enhancing the details and shading to give it a more realistic appearance. The artist then switches to a yellow pen and begins coloring the center of the flower, adding depth and contrast to the drawing. The video concludes with the artist adding final touches to the coloring, enhancing the overall appearance of the flower illustration. |
| User | What could this flower be? |
| LLaVA-Video | This flower could be a magnolia or a similar species. |

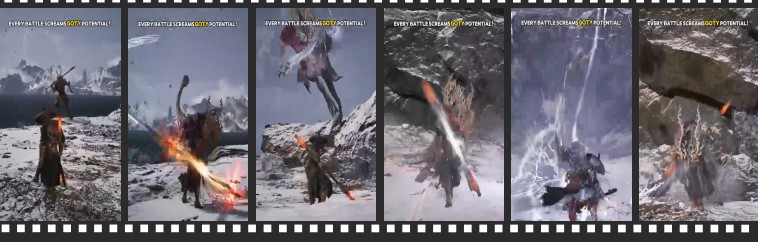

| User | Please describe this video. |
|---|---|
| LLaVA-Video | The video showcases an intense battle scene set in a snowy, mountainous landscape. A character dressed in dark, heavy armor with glowing orange accents wields a sword and faces off against a large, menacing creature with glowing red eyes and antlers. The text 'EVERY BATTLE SCREAMS GOTY POTENTIAL!' is prominently displayed at the top of each frame, emphasizing the high stakes and epic nature of the confrontation. The battle is fierce, with the character and the creature exchanging powerful attacks. The creature emits blue energy and lightning, while the character retaliates with fiery and icy attacks. The snowy ground is littered with debris and frost, adding to the harsh and brutal environment. The background features a large body of water and a distant lighthouse, enhancing the sense of isolation and danger. |

Table 11: LLaVA-Video learns to understand the the video in *special domain*.

**Real-world Knowledge: Unusual Action**

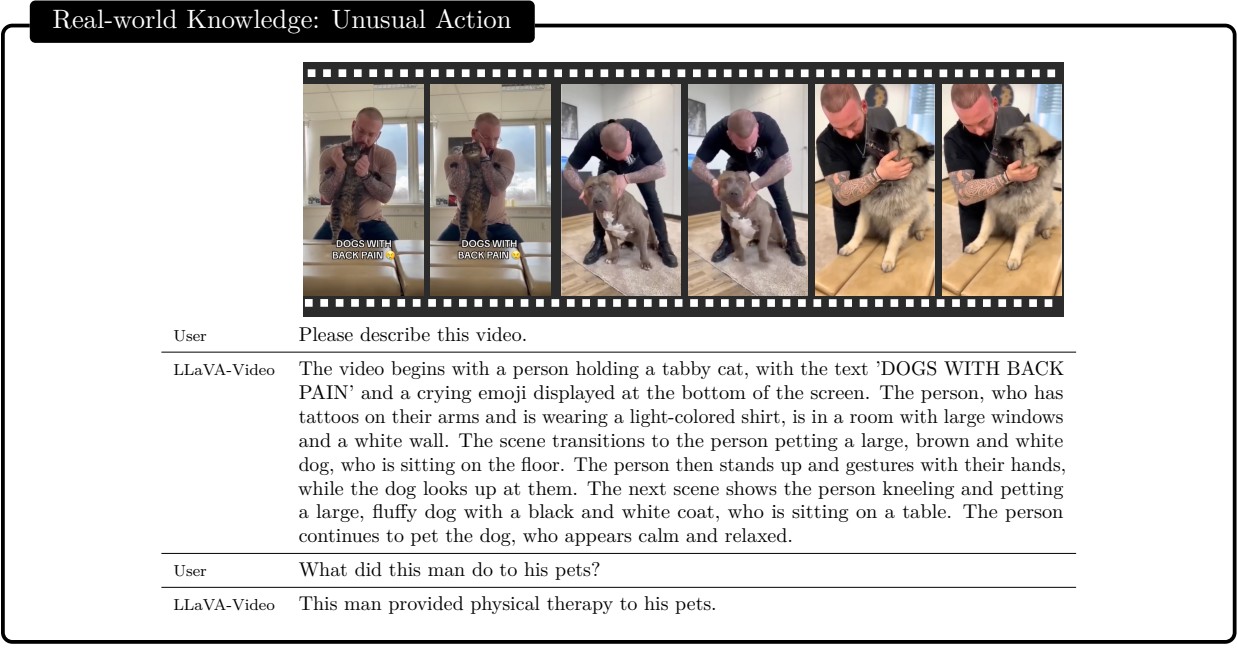

| User | Please describe this video. |
|---|---|
| LLaVA-Video | The video begins with a person holding a tabby cat, with the text 'DOGS WITH BACK PAIN' and a crying emoji displayed at the bottom of the screen. The person, who has tattoos on their arms and is wearing a light-colored shirt, is in a room with large windows and a white wall. The scene transitions to the person petting a large, brown and white dog, who is sitting on the floor. The person then stands up and gestures with their hands, while the dog looks up at them. The next scene shows the person kneeling and petting a large, fluffy dog with a black and white coat, who is sitting on a table. The person continues to pet the dog, who appears calm and relaxed. |
| User | What did this man do to his pets? |
| LLaVA-Video | This man provided physical therapy to his pets. |

Table 12: LLaVA-Video learns to understand the *unusual action* in the video.

**Real-world Knowledge: Physical Laws**

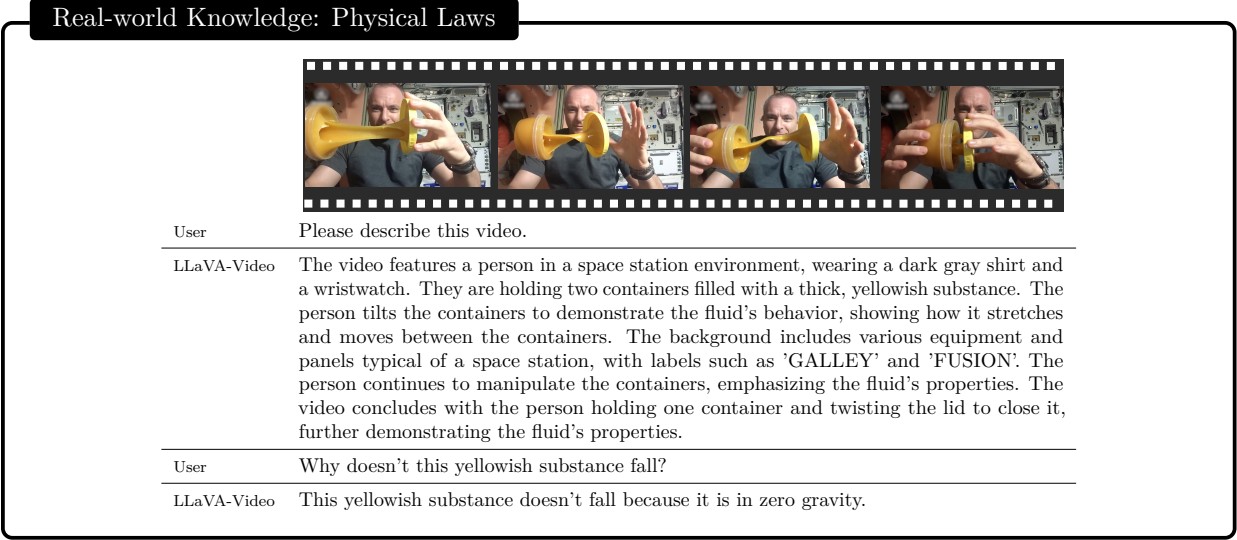

| User | Please describe this video. |
|---|---|
| LLaVA-Video | The video features a person in a space station environment, wearing a dark gray shirt and a wristwatch. They are holding two containers filled with a thick, yellowish substance. The person tilts the containers to demonstrate the fluid's behavior, showing how it stretches and moves between the containers. The background includes various equipment and panels typical of a space station, with labels such as 'GALLEY' and 'FUSION'. The person continues to manipulate the containers, emphasizing the fluid's properties. The video concludes with the person holding one container and twisting the lid to close it, further demonstrating the fluid's properties. |
| User | Why doesn't this yellowish substance fall? |
| LLaVA-Video | This yellowish substance doesn't fall because it is in zero gravity. |

Table 13: LLaVA-Video learns to understand the *physical laws* in the video.

