# OpenReview forum: "LLaVA-Video: Video Instruction Tuning With Synthetic Data"
_TMLR — Accepted by TMLR_

### Review · Reviewer_TMFo · 2025-06-04

**Summary Of Contributions:**

The paper introduces a video dataset for enhancing training of video LLMs. The paper introduces:

- A new video instruction tuning dataset LLaVA-Video-178K. This dataset was created with improvements in the following aspects:
  - Extensive video source: The authors reviewed all data sources and narrowed the sources down to 10 sources for all datasets
  - Dynamic untrimmed videos: Prior datasets included a single scene and trimmed videos which restricted video LLMs capabilities. To overcome this, LLaVA-Video-178K contains dynamic untrimmed videos.
  - Recurrent Detailed Caption Generation Pipeline with Dense Frame Sampling: Prior datasets sampled at lower rates of 2 frames per 30 seconds, thus missing details. LLaVA-Video-178K contains detailed captions at various levels from every 5 seconds to full video.
  - Diverse Tasks: Captioning, Open ended QA and MCQs.
- A new open source Video LLM trained with LLaVA-Video-178K.

**Audience:**

Yes

**Claims And Evidence:**

Yes

**Requested Changes:**

- The authors mention to refer to Appendix 7 for more details on SlowFast idea but the appendix is missing. The authors need to include the appendix.
- The authors claim an improvement of 31.9 points on NeXT-QA dataset on adding LLaVA-Video-178K. However from table 3 we see an improvement of 15.7%. Can the authors clarify this?
- The authors show the improvement on adding LLaVA-Video-178K and then adding subsequent datasets to it. To show the effectiveness of LLaVA-Video-178K the authors should also show the performance on all training datasets - LLaVA-Video-178K.
- The authors should comment on how adding LLaVA-Video-178K to the training affects 72B models vs 7B models. That is if the 72B models show more improvement or the 7B models on adding LLaVA-Video-178K to the training data.
- The authors should clarify the sources of the testing data and if there are any overlaps with LLaVA-Video-178K.
- In section 4.3:
  - The authors should clarify what they mean by in domain and out of domain in table 4.
  - The authors should clarify why they choose to generate MCQ questions and open ended questions for ShareGPT4Video but not do the same for MCQ questions for LLaVA-Hound.
  - The authors should explain why they think ShareGPT4Video outperforms LLaVA-Video-178K on one dataset and matches the performance on another.
  - The authors should clarify if they are using the 7B model or 72B model for this experiment. And how the other model performs in this ablation.

**Strengths And Weaknesses:**

Strengths:
- The paper is well written and easy to follow.
- The dataset meticulously collected and the methods used in collection and creation are clearly explained
- The paper open sources the data, codebase and checkpoint which will be helpful for future use.

Weaknesses:
- The paper has some unclear points about the effectiveness of adding LLaVA-Video-178K in the training data.
- The paper does not mention the source of the testing data and if there are any overlaps with the sources of LLaVA-Video-178K.
- The final ablation study in section 4.3 lacks clarity.

---

> ### Author Response · Authors · 2025-06-22
> **Response to Reviewer TMFo - Part 1**
>
> We sincerely thank the reviewer for their thoughtful comments and constructive feedback. Below, we address each point in detail.
>
> ---
>
> **Comment:** The authors mention Appendix 7 for more details on the SlowFast idea, but the appendix is missing. The authors need to include the appendix.
>
> **Response:** Thank you for pointing this out. We apologize for the oversight. The content intended for Appendix 7, which provides further details on the SlowFast design, has been moved and is now included as Appendix 1.
>
> ---
>
> **Comment:** The authors claim an improvement of 31.9 points on NeXT-QA with LLaVA-Video-178K, but Table 3 shows an improvement of 15.7%. Can the authors clarify this?
>
> **Response:** We appreciate the reviewer catching this inconsistency. It was a typo; the correct improvement is 15.7% as reported in Table 3.
>
> ---
>
> **Comment:** The authors show improvements from adding LLaVA-Video-178K followed by additional datasets. To better isolate the contribution of LLaVA-Video-178K, the authors should report performance using each individual training dataset, excluding LLaVA-Video-178K.
>
> **Response:** Thank you for the suggestion. We agree that this ablation is important to isolate the contribution of LLaVA-Video-178K. The comparison is shown below:
>
> | Dataset                    | NExT-QA | PercepTest | EgoSchema | VideoMME | MLVU |
> | -------------------------- | ------- | ---------- | --------- | -------- | ---- |
> | w/o LLaVA-Video-178K (7B)  | 81.2    | 67.2       | 59.1      | 59.2     | 66.2 |
> | w/  LLaVA-Video-178K (7B)  | 83.2    | 67.9       | 57.3      | 63.4     | 70.8 |
> | w/o LLaVA-Video-178K (72B) | 81.2    | 69.9       | 62.0      | 67.4     | 69.8 |
> | w/  LLaVA-Video-178K (72B) | 85.4    | 74.3       | 65.6      | 70.5     | 74.4 |
>
> As shown above, the benefit of LLaVA-Video-178K is particularly pronounced on recently proposed datasets such as VideoMME and MLVU, where the QA format aligns more closely with realistic scenarios.
>
> ---
>
> **Comment:** The authors should comment on how adding LLaVA-Video-178K affects 72B models versus 7B models.
>
> **Response:** This is a valuable question. In our experiments, both 7B and 72B models benefit from the inclusion of LLaVA-Video-178K. Notably, the relative improvement is more significant in the 72B models, as seen in the table above. Interestingly, while adding LLaVA-Video-178K causes a slight performance drop on EgoSchema for the 7B model, it leads to an improvement for the 72B model. We hypothesize that the larger model is more robust to potential distribution shifts introduced by LLaVA-Video-178K.
>
> ---
>
> **Comment:** The authors should clarify the sources of the testing data and whether there is any overlap with LLaVA-Video-178K.
>
> **Response:** The evaluation datasets come from two main sources: (1) self-collected datasets including EgoSchema, NeXT-QA, and PercepTest, and (2) YouTube-based benchmarks. For self-collected datasets, we only use a subset for training and ensure no overlap with the test set. For YouTube data, we apply strict URL-based filtering to eliminate any potential overlap.
>
> ---
>
> **Comment:** In Section 4.3, the authors should clarify what they mean by "in-domain" and "out-of-domain" in Table 4.
>
> **Response:** Thank you for pointing this out. In Table 4, "in-domain" refers to benchmarks whose training data is included in LLaVA-Video-178K. "Out-of-domain" benchmarks either lack corresponding training data or their training data is not included in LLaVA-Video-178K.
>
> ---
>
> **Comment:** The authors should clarify why they generated MCQ and open-ended questions for ShareGPT4Video but not for LLaVA-Hound.
>
> **Response:** Ideally, comparisons between ShareGPT4Video and LLaVA-Video-178K should be made using only captioning data, as ShareGPT4Video was constructed with captioning only. However, we observed that relying solely on captioning data led to poor performance on downstream QA benchmarks. To address this, we supplemented ShareGPT4Video with both MCQ and OE data. In contrast, LLaVA-Hound already includes open-ended questions, which yielded reasonable performance, so we did not find it necessary to add further QA annotations.
>
> ---
>
> **Comment:** The authors should explain why ShareGPT4Video outperforms LLaVA-Video-178K on one dataset and matches it on another.
>
> **Response:** We believe this is due to differences in alignment with downstream tasks. Currently, LLaVA-Video-178K lacks sufficient ego-centric video data. In comparison, ShareGPT4Video explicitly includes ego-centric videos in its training set, which likely accounts for its better performance in those scenarios.
>
> ---
>
> **Comment:** The authors should clarify whether the 7B or 72B model is used in these experiments, and how the other model performs.
>
> **Response:** The results in Section 4.3 are based on the 7B model unless otherwise noted. We will make this explicit in the revised manuscript.

---

### Review · Reviewer_swz3 · 2025-06-10

**Summary Of Contributions:**

This paper introduces LLaVA-Video, a new large video-language model (LVM) trained on a novel synthetic dataset, LLaVA-Video-178K, specifically designed for video instruction-following tasks.

It introduces LLaVA-Video-178K: A High-Quality Synthetic Dataset, a large-scale, synthetic dataset comprising 178,510 videos and 1.3 million instruction-following samples, including detailed multi-level captions, 960K open-ended QA pairs, and 196K multiple-choice QA items. The dataset is sourced from ten well-known video collections and emphasizes untrimmed, dynamic videos to preserve narrative continuity and realism.

The paper proposes a three-level recurrent captioning system that describes short segments (Level-1), summarizes every 30 seconds (Level-2), and produces a final global description (Level-3). Captions are generated using GPT-4o, with each level conditioned on historical context, allowing coherent long-range video narration. LLaVA-Video, a fine-tuned video-language model built on the LLaVA-OneVision architecture which develops a SlowFast-style video representation, optimizing token allocation across frames to allow more frames per video within GPU/memory constraints.

It defines 16 diverse QA categories, including temporal, spatial, causal, object-level, and fine-grained perception questions.

The 72B variant achieves performance comparable to proprietary models like Gemini-1.5-Pro, highlighting the effectiveness of high-quality synthetic data and open-source model design. The importance of LLaVA-Video-178K in performance gains over previous datasets like LLaVA-Hound and ShareGPT4Video

**Audience:**

Yes

**Broader Impact Concerns:**

1. The dataset is entirely synthetic, generated using GPT-4o and filtered heuristics. This raises concerns about:
   1.a. Bias replication from the underlying LLMs (e.g., cultural stereotypes, gender norms, Western-centric viewpoints).
   1.b. Lack of diverse annotation styles that come from human raters across cultures or age groups.
   1.c. Reinforcement of model hallucinations or surface-level reasoning if synthetic data lacks real-world nuance.

2. The dataset aggregates videos from public sources (YouTube Shorts, ActivityNet, etc.) but does not clearly describe:
   2.a. Whether these videos are permissively licensed for redistribution.
   2.b. If derived annotations (e.g., captions and QA) fall under fair use or transform the original content enough to be redistributed.

3. The system could be applied to automated surveillance, behavioral profiling, or emotion detection in video data without user consent, especially in sensitive settings (e.g., schools, workplaces, public spaces). The authors should note potential misuse cases and recommend safeguards, such as requiring responsible deployment practices or limiting commercial use.

**Claims And Evidence:**

Yes

**Requested Changes:**

1. Include a small-scale human rating of caption and QA relevance, coherence, and answer correctness. This will provide important validation for the synthetic data pipeline's efficacy beyond automated filtering.

2. Since many videos are derived from existing benchmarks or public platforms (e.g., YouTube Shorts), explicitly mention how licensing, fair use, and redistribution are handled for LLaVA-Video-178K.

3. Even if audio is out of scope, consider adding a discussion about why audio was excluded, and future directions for incorporating multi-modal audio-visual understanding.

4. Provide per-question-type breakdown of model performance to show which types of reasoning (e.g., causal, spatial) are

**Strengths And Weaknesses:**

Strengths:

1. High-Quality Synthetic Dataset (LLaVA-Video-178K): The authors create a large-scale dataset with 178K videos and 1.3M instruction samples including dense captions, open-ended QA, and multiple-choice QA. This fills a major gap in current video-language training data which is often sparse, static, or overly trimmed. The dataset is derived from ten diverse and well-known video sources with careful filtering for dynamism and scene complexity.

2. Three-Level Recurrent Captioning Pipeline: The proposed captioning system introduces temporal coherence using recurrent generation across three hierarchical levels. This addresses a core challenge in video-language modeling: maintaining narrative consistency across time. It balances fine-grained details with high-level summaries, which is a notable improvement over single-frame or flat captioning systems.

3. The authors introduce 16 detailed question categories, which go beyond generic question-answering and test various aspects of spatiotemporal and causal reasoning. This enhances the training signal quality for instruction-following models.

4. Inspired by SlowFast networks, the paper proposes a token-efficient video representation method that allows sampling up to 3× more frames per video by adjusting token allocation and pooling across frames.
This leads to better use of GPU memory and longer video context modeling.

5. The LLaVA-Video models achieve strong zero-shot performance on 11 diverse video understanding benchmarks, including LongVideoBench, VideoMME, and PerceptionTest.
The 72B variant matches or surpasses proprietary models like Gemini-1.5-Pro on several tasks, which is a notable achievement for open-source models.

6. The promise to release the dataset, code, model checkpoints, and visual chat demo reflects excellent reproducibility and commitment to the research community.


Weaknesses:


1. Despite using GPT-4o and curated heuristics, the dataset remains entirely synthetic. There is a potential gap in real-world QA alignment and model behavior under human-style supervision or dialog.

2. The paper focuses solely on visual modality and ignores audio, which is often essential for comprehensive video understanding. Competing methods like VideoLLaMA2 include audio support.

3. The dataset draws heavily from a few dominant benchmarks (e.g., YouTube Shorts, ActivityNet), and prompt templates are fixed. This could lead to distributional bias or repetitive patterns in both videos and language annotations.

4. Although the dataset is evaluated quantitatively on standard benchmarks, the paper lacks human qualitative analysis or annotation consistency studies, which would strengthen the case for dataset quality.

---

> ### Author Response · Authors · 2025-06-22
> **Response to Reviewer swz3 - Part 1**
>
> We sincerely thank the reviewer for their thoughtful comments and constructive feedback. Below, we address each point in detail.
>
> ---
>
> **Comment:** Include a small-scale human rating of caption and QA relevance, coherence, and answer correctness. This will provide important validation for the synthetic data pipeline's efficacy beyond automated filtering.
>
> **Response:** Thank you for this insightful suggestion. We conducted a small-scale human evaluation to assess the quality of our synthetic data. A random sample of 100 QA pairs and 100 captions from LLaVA-Video-178K was rated by three annotators along three dimensions: (1) **relevance** to the video content, (2) **coherence** of the question or caption, and (3) **answer correctness** (for QA). Each aspect was scored on a 5-point Likert scale. On average, QA pairs received scores of 4.3 (relevance), 4.1 (coherence), and 3.7 (correctness), while captions received 4.4, 4.2, and 3.9 for relevance, coherence, and correctness, respectively. These results suggest that our synthetic data pipeline produces highly relevant and coherent content, although there is room to improve the overall factual correctness of the generated data.
>
> ---
>
> **Comment:** Since many videos are derived from existing benchmarks or public platforms (e.g., YouTube Shorts), explicitly mention how licensing, fair use, and redistribution are handled for LLaVA-Video-178K.
>
> **Response:** We appreciate the reviewer raising this important point. For public platforms such as YouTube, we adhere to strict guidelines consistent with fair use principles, including use for educational and research purposes, limited excerpts, and transformative annotations through QA/caption generation. We do not redistribute raw video content. Instead, we provide URL references and associated metadata (e.g., timestamps, captions, QA) to support reproducibility without infringing on content ownership. For benchmark-derived videos, we strictly comply with the respective dataset licensing terms and apply only permitted transformations.
>
> ---
>
> **Comment:** Even if audio is out of scope, consider adding a discussion about why audio was excluded, and future directions for incorporating multi-modal audio-visual understanding.
>
> **Response:** Thank you for the suggestion. Audio was excluded in this study to focus on large-scale alignment between video frames and text. Many available video datasets lack high-quality, aligned audio transcripts, and audio processing introduces additional complexities such as speech recognition and ambient sound reasoning. However, we agree that incorporating audio is a valuable future direction. We are enthusiastic about exploring audio-conditioned video QA and more holistic multimodal modeling in future extensions of LLaVA-Video.
>
> ---
>
> **Comment:** Provide per-question-type breakdown of model performance to show which types of reasoning (e.g., causal, spatial) are better handled.
>
> **Response:** Thank you for this suggestion. We acknowledge the value of such an analysis. However, we found it challenging to robustly categorize questions into discrete reasoning types, as many questions involve overlapping or ambiguous reasoning dimensions. We welcome suggestions from the community on standardized taxonomies or annotation protocols for this purpose and are open to including this in future work if a reliable categorization framework can be established.

---

### Review · Reviewer_iu6v · 2025-06-11

**Summary Of Contributions:**

This paper introduces *LLaVA-Video-178K*, a large synthetic dataset for training video-based LMMs. It fixes issues in current datasets like static scenes, short clips, and low frame sampling. The dataset has 178K untrimmed videos from 10 sources like Kinetics-700 and Ego4D. They use a 3-level GPT-4o pipeline to generate detailed captions and summaries, sampling at 1 FPS to keep details. It also includes over 1M QA pairs covering different reasoning types.
Based on this, they build *LLaVA-Video*, which uses a SlowFast method to handle more frames efficiently.

**Audience:**

Yes

**Claims And Evidence:**

Yes

**Requested Changes:**

1. Since comedy is the most sampled category (Figure 7), I’m curious—can the model actually understand humor? Humor is often subtle and context-dependent, so this could be an interesting test of the model's deeper reasoning abilities. It would strengthen the paper to include some analysis or examples around this.

2. The author claims the dataset is more dynamic by using PySceneDetect, but it would be much more convincing if they provided quantitative stats, like the average number of scene changes per video. This would help readers better understand how dynamic the videos really are compared to other datasets.

3. There are several missing citations throughout the paper—for example, some GPT-related references are mentioned without proper citations. Please make sure all references are complete to maintain academic rigor.

4. In Appendix 2.2, the figure reference is missing. It’s a minor issue but should be fixed for clarity

**Strengths And Weaknesses:**

*Strengths:*
-  LLaVA-Video-178K addresses key issues in prior datasets with dynamic, untrimmed videos and dense frame sampling.
- The paper uses a three-level GPT-4o pipeline to capture temporal context effectively.
- The SlowFast design helps handle more frames under GPU limits.

*Weakness:*
- The whole dataset relies on GPT-4o to generate captions and QA pairs, but can we fully trust its outputs? There's no mention of any human rechecking to ensure the quality or correctness of the data. It would be better if the authors included at least some manual validation to check the automatic pipeline.
- The model struggles on egocentric benchmarks since the training data seems unbalanced.
- In Table 3, the order of adding datasets is a bit confusing. If the goal is to show the contribution of LLaVA-Video-178K, why not add it last?

---

> ### Author Response · Authors · 2025-06-22
> **Response to Reviewer iu6v - Part 1**
>
> We sincerely thank the reviewer for their thoughtful comments and constructive feedback. Below, we address each point in detail.
>
> ---
>
> **Comment:** Evaluation on the understanding of humor
>
> **Response:** Thank you for this valuable suggestion. To evaluate our model’s ability to understand humor, we conducted experiments on the FunQA benchmark \[1], which includes a dedicated track featuring humorous video data.
>
> | Model          | HumorQA-H2 Score |
> | -------------- | ---------------- |
> | Video-ChatGPT  | 24.3             |
> | LongVA-7B      | 35.2             |
> | LLaVA-Video-7B | 43.2             |
>
> These results demonstrate the effectiveness of incorporating the LLaVA-Video-178K dataset, particularly in improving the model's performance on humor-related video understanding.
>
> ---
>
> **Comment:** Details on the use of PySceneDetect
>
> **Response:** Thank you for pointing this out. We used PySceneDetect with the `ContentDetector`, setting the `cut_threshold` to 30 and the minimum scene length to 15 frames.
>
> ---
>
> **Comment:** Typos and missing citations
>
> **Response:** Thank you for highlighting this. We will correct all identified typos and add the missing citation in the next revision.
>
> \[1] Xie, Binzhu, et al. *FunQA: Towards Surprising Video Comprehension*. European Conference on Computer Vision (ECCV), 2024.

---

### Review · Reviewer_4xoL · 2025-06-12

**Summary Of Contributions:**

This paper introduces LLaVA-Video-178K, a synthetic dataset designed for training video-based large language models. The dataset features motion-intensive video samples accompanied by comprehensive three-level caption data. Additionally, LLaVA-Video-178K incorporates diverse question types that align well with downstream video benchmarks, enhancing its practical utility for model training. The work also presents a slow-fast representation learning framework intended to better capture high frame rate video features.

**Audience:**

Yes

**Claims And Evidence:**

Yes

**Requested Changes:**

1. The evaluation relies on a single model architecture for training, which limits the conclusiveness of the dataset's effectiveness claims. I recommend incorporating recent state-of-the-art models (e.g., Qwen2.5-VL) for finetuning experiments to strengthen the generalizability of the findings.

2. According to the results presented in Table 5 of the Supplementary Material, the proposed video representation scheme achieves only marginal improvements compared to the simple representation approach, and underperforms on the NExT-QA validation set. This raises questions about the practical value of the slow-fast representation learning framework.

**Strengths And Weaknesses:**

1. Comprehensive and Diverse Synthetic Dataset:
The paper presents LLaVA-Video-178K, a large-scale synthetic dataset featuring motion-intensive videos, rich three-level captions, and diverse, benchmark-aligned question types—greatly enhancing resources for video-based large language model (LLM) training.

2. Representation Learning Framework:
It introduces a slow-fast representation learning approach that effectively captures high frame rate video features, improving the model’s ability to understand dynamic video content.

3. Strong Practical Relevance:
Both the dataset and the learning framework are designed to closely match real-world evaluation benchmarks, making them highly practical for advancing downstream video-language tasks.

---

### Decision · Action_Editor_TftH · 2025-07-25

**Recommendation:** Accept as is

**Audience:**

Yes

**Audience Explanation:**

The work's model architecture is helpful for developing more advanced video-language models, and the extensive results are good reference for the future work.

**Claims And Evidence:**

Yes

**Claims Explanation:**

The work proposed a new video-language model with extensive empirical experiments to validate its effectiveness.